# Chloride Intracellular Channel Protein 2 Promotes Microglial Invasion: A Link to Microgliosis in the Parkinson’s Disease Brain

**DOI:** 10.3390/brainsci13010055

**Published:** 2022-12-28

**Authors:** Mohammed E. Choudhury, Saya Ozaki, Noriyuki Miyaue, Taisei Matsuura, Kanta Mikami, Afsana Islam, Madoka Kubo, Rina Ando, Hajime Yano, Takeharu Kunieda, Masahiro Nagai, Junya Tanaka

**Affiliations:** 1Department of Molecular and Cellular Physiology, Graduate School of Medicine, Ehime University, 454 Shitsukawa, Toon, Ehime 791-0295, Japan; 2Department of Neurosurgery, Graduate School of Medicine, Ehime University, 454 Shitsukawa, Toon, Ehime 791-0295, Japan; 3Department of Clinical Pharmacology and Therapeutics, Graduate School of Medicine, Ehime University, 454 Shitsukawa, Toon, Ehime 791-0295, Japan

**Keywords:** microglia, CLIC2, MMP-9, Parkinson’s disease

## Abstract

Activated microglia potentially cause neurodegeneration in Parkinson’s disease (PD). Matrix metalloproteinase (MMP)-9 plays a crucial role in the pathogenesis of PD, but the modulator of microglial release of MMP-9 remains obscure. Given the modulatory effect of chloride intracellular channel protein 2 (CLIC2) on MMPs, we aimed to determine the role of CLIC2 in regulating microglial MMP expression and activation. We found that CLIC2 is expressed in microglia and neurons in rat brain tissue and focused on the function of CLIC2 in primary cultured microglia. Exposure to recombinant CLIC2 protein enhanced microglial invasion activity, and its knockdown abolished this activity. Moreover, increased activation of MMP-9 was confirmed by the addition of the CLIC2 protein, and CLIC2 knockdown eliminated this activation. Additionally, increased expression of CLIC2 was observed in PD-modeled tissue. In conclusion, CLIC2 increases MMP-9 activity in the microglia, which are involved in PD pathogenesis.

## 1. Introduction

The accumulation of microglia in lesioned areas is considered a dominant factor in Parkinson’s disease (PD) pathophysiology because they are potentially harmful to neurons when gathered [1]. The mechanisms and molecules responsible for microglial invasion in degenerated neuronal areas have drawn considerable attention in PD research. Several animal studies have suggested that extracellular matrix metalloproteinases (MMPs) could be major players in the progression of PD. In particular, genetic deletion and pharmacological inhibition of MMP-9 remarkably reduced microglial activation and dopaminergic neuronal degeneration in a 1-methyl-4-phenyl-1,2,3,6-tetrahydropyridine mouse model of PD [2,3]. MMPs have favorable roles in the CNS, including the facilitation of tissue repair and synaptic plasticity [4,5].

To identify the molecule that increased MMP-9 expression, we prepared a 6-hydroxydopamine (6-OHDA)-induced rat model of PD [6], where we found remarkably increased mRNA expression of MMP-9 and chloride intracellular channel protein 2 (CLIC2). CLICs have been identified as chloride ion channel proteins, but a considerable number of these proteins are present in the soluble fractions of the cytosol and nucleus [7]. CLIC1–6 proteins have been implicated in membrane remodeling, intracellular trafficking, vacuole formation, and cytoskeleton reorganization [8]. The pharmacological and genetic inhibition of CLICs abolished NLRP3 inflammasome activation by suppressing the potassium efflux–mitochondrial reactive oxygen species axis in macrophages [9]. In the mouse brain, immunoreactivity of CLIC1 was found in the cytosol of ramified microglia, where it plays a crucial role in microglial activation and the generation of reactive oxygen species [10]. The cell-type expression pattern of CLIC4-GFP in mouse brains confirmed the robust expression of CLIC4 on microglial cells, which regulates the extracellular matrix degradation activity of MMP14 [11]. In addition, marked expression of CLIC2 has been confirmed in myeloid cells of human colon tissue in our earlier report [12], but its function in myeloid cells is unknown. In the present study, we used primary cultured microglia from rats, in which recombinant CLIC2 and CLIC2-siRNA were employed to uncover the role of CLIC2 in microglial immunosurveillance.

## 2. Materials and Methods

### 2.1. Animals

All animal experiments were conducted in accordance with the Guidelines of the Ethics Committee for Animal Experimentation of Ehime University. Male rats were housed in a controlled animal facility in standard cages with a 12-h light/dark cycle (lights on at 7:00; lights off at 19:00) and a temperature of 25 °C. To prepare the 6-OHDA-induced PD model, animals were maintained under isoflurane anesthesia and placed in a stereotactic instrument (Narishige, Tokyo, Japan). 6-OHDA (Toronto Research Chemicals, Inc., Toronto, ON, Canada) was prepared by dissolving it in saline containing ascorbic acid (Wako, Osaka, Japan) (10 μg/μL dissolved in 1% ascorbate saline). Unilateral injection of 6-OHDA was given to the rats; 4 μL of 6-OHDA was injected using a Hamilton syringe w (26-gauge needle) in the vicinity of the right medial forebrain bundle at anteroposterior (AP) −4.5 mm, mediolateral (ML) +1.5 mm, and dorsoventral (DV) −8.0 mm, with an injection flowrate speed of 1 μL/min. The needle was slowly withdrawn 5 min after the injection. The skin on the head of the rats was sutured with surgical needle-equipped sutures (Alfresa Pharma Corporation, Osaka, Japan).

### 2.2. Cell Culture

Primary cultures of microglia were prepared from 1–3-day-old neonatal rats. Briefly, after removing the whole forebrain from neonatal rats, a nylon bag with 160 µm pores was used to dissociate into individual cells. Dissociated cells were cultured in 75 cm^2^ flasks with 10% fetal calf serum (FCS)-supplemented Dulbecco’s modified Eagle’s medium (DMEM; Wako, Osaka, Japan) as mixed glial cultures. After 11 days, microglial cells were obtained from the culture. To do this, the flasks were agitated at 200 rpm for 1 h, and after that, the cells were seeded on 12- or 24-well poly-L-lysine-coated plates and incubated with E2 medium (serum-free DMEM containing 10 mM HEPES, pH 7.3, (Gibco, Grand Island, NY, USA), 4.5 mg/mL glucose, 5 µg/mL insulin, 5 nM sodium selenite, 5 µg/mL transferrin (Gibco), and 0.1 mg/mL bovine serum albumin (Merck) for 1 h. The CLIC2 recombinant protein was prepared according to a cell-free protein synthesis system using wheat germ ribosomal RNA [13].

CLIC2 knockdown was performed using RNA interference as described previously [13]. The siRNAs targeting the CLIC2 gene were as follows: 5′-CCAUUCUUGAUAUAUAACA-3′ and 5′-UGUUAUAUAUCAAGAAUGG-3′ (Sigma-Aldrich, St. Louis, MO, USA). An siRNA duplex with an irrelevant sequence (IRR-siRNA; 5′-GCGCGCUUUGUAGGAUUCGTT-3′ and 5′-CGAAUCCUACAAAGCGCGCTT-3′; Dharmacon Research, Pittsburgh, PA, USA) was used as a control. To prepare the knockdown cell, the cells were incubated for 24 h with the siRNA following culturing in fresh culture medium for 48 h. The accuracy of the knockdown was confirmed with immunoblotting [14].

### 2.3. Immunoblotting

Cells were homogenized with Laemmli’s sample solution containing 3% sodium dodecyl sulfate to electrophoresis and transfer. The lysates were immunoblotted with antibodies to β-actin monoclinal antibody (Wako) and CLIC2 polyclonal antibody (Abcam, Cambridge, UK). The blots were visualized with alkaline phosphatase-labeled secondary antibodies (Promega, Madison, WI, USA) and evaluated by densitometry using ImageJ 1.43u (Wayne Rasband) [15].

### 2.4. Quantitative Real-Time RT-PCR (qPCR)

To extract RNA from microglia, cells were homogenized with QIAshredder, and RNA was purified using the RNeasy Mini Kit (QIAGEN, Venlo, The Netherlands). ReverTra Ace qPCR RT Master Mix (TOYOBO, Osaka, Japan) was used for cDNA synthesis, and THUNDERBIRD^®^ Next SYBR^®^ qPCR Mix (TOYOBO) was used for qPCR. The sequences of primers used in this study were as follows: MMP-2 forward (TGGACTCTAGGAGAAGGACAA) and reverse (CTGCTGTATTCCCGACCATAA); MMP-9, forward (CTCTGCCTGCACCACTAAA) and reverse (TCGAGTAGGACAGAAGCCATA); CLIC2, forward (GCTGGAAGTGACGGAGAAAG) and reverse (TACCTTGGAGGAGCGAGTGT); and GAPDH, forward (GAGACAGCCGCATCTTCTTG) and reverse (TGACTGTGCCGTTGAACTTG). Glyceraldehyde 3-phosphate dehydrogenase (GAPDH) was used as a housekeeping (reference) gene, and all gene-specific mRNA expression values were presented as relative expression levels normalized to GAPDH. Gene expression was quantified using ΔCt values, where ΔCt is expressed as the difference between the target and reference gene Ct values [16].

### 2.5. Gelatin Zymography

The supernatant of the treated microglial cell culture was collected after the indicated time period and concentrated using an Amicon^®^ Ultra-4 (ultracel-3K) centrifugal filter device (Merck Millipore Ltd., Bedford, MA, USA). The collected supernatant was centrifuged at 7500× *g* for 50 min at 4 °C. The protein content of the concentrated media was quantified using Pierce BCA Protein Assay Kit (Thermo Fisher Scientific Inc., Waltham, MA, USA). The activity of MMP-9 was analyzed in the cell supernatants by gelatin zymography using a Gelatin-zymography Kit (Cosmo Bio Co., Tokyo, Japan) according to the manufacturer’s instructions. Briefly, all samples with equal protein content were mixed with sample buffer and incubated for 15 min at RT. The samples and markers were then loaded onto gelatin-gel plates for electrophoresis. The gel was washed and incubated for 40 h in reaction buffer at 37 °C. After the enzymatic reaction, the gel was stained with Coomassie Blue and incubated for 30 min at RT. The intensity of the bands was determined using ImageJ. Gelatin zymography was performed three times [13].

### 2.6. Transmigration Assay

BioCoat™ Matrigel^®^ Invasion Chambers with 8.0 µm PET Membrane (Corning, NY, USA) were used for the in vitro assay to assess the invasive activity of primary cultured microglial cells. Briefly, microglial cells, both siRNA- and peptide-treated or untreated, were suspended in DMEM containing 0.5% BSA and seeded into the upper chambers at a density of 5 × 10^4^ cells/insert. The lower chamber of the Falcon 24-well plates was filled with 500 μL of DMEM containing 1% FBS. The cells were incubated for 24 h, after which the cells on the upper membrane surface were removed mechanically. Cells on the lower side of the membrane were fixed, stained with 0.1% crystal violet, and observed under a microscope (×2) to count the number of cells or measure the area covered with cells [13].

### 2.7. Metabolic Flux Analyses

The primary cultured microglial oxygen consumption rate (OCR) was measured using a Seahorse XFp Flux Analyzer (Agilent Technologies, Santa Clara, CA, USA), where a Seahorse XFp Cell Mito Stress Test Kit (Agilent Technologies) was used. Microglial cells were seeded at a density of 2.5 × 10^4^ cells in eight-well plates (Agilent Technologies) and incubated with or without recombinant CLIC2 peptides overnight. After incubation, the assay was performed according to the manufacturer’s instructions [17].

### 2.8. Immunohistochemical Staining

CO_2_-induced euthanized rats were transcardially perfused with 4% paraformaldehyde containing 2 mM MgCl_2_ for 10 min at a flow rate of 60 mL/min to dissect the brains. The dissected brains were immersed in 15% sucrose in PBS at 4 °C overnight and rapidly frozen in dry ice. After that, the brain was sliced into 10 μm thick coronal sections at the regions of substantia nigra par compacta. Brain sections were incubated with primary antibodies against Mouse monoclonal CD11b (GentexIrvine, Irvine, CA, USA), Rabbit polyclonal CLIC2 (Abcam, Cambridge, UK), and Sheep polyclonal TH (Novus Biologicals, Littleton, CO, USA), followed by incubation with DyLight 488-, DyLight 549-, and DyLight 649-labeled secondary antibodies (Jackson ImmunoResearch Laboratories, West Grove, PA, USA). Hoechst 33258 (Sigma-Aldrich) was used for the nuclear staining. Immunostained specimens were observed with BZ-X800 fluorescence microscope (Keyence, Osaka, Japan) using a 40× objective lens, and morphometric analysis were performed as described in our earlier report [15].

### 2.9. High-Performance Liquid Chromatography

High-performance liquid chromatography (HPLC) was employed to determine the levels of dopamine (DA) in the striatal tissue, as previously described [6]. Briefly, lysates were prepared from tissues that had been homogenized with an ultrasonic cell disruptor in 0.1 M perchloric acid containing 5 mM EDTA and 3,4-dihydroxybenzamine (Wako), and they were instilled into an HPLC apparatus with a reversed-phase column. Dopamine concentrations were quantified using synthetic DA (Wako) as an external standard and 3,4-dihydroxybenzamine as an internal standard.

### 2.10. Statistics

Data were expressed as the mean ± SD. Group means were compared using a two-tailed unpaired *t*-test with a non-parametric approach and two-way ANOVA with Tukey’s multiple comparison test. All analyses were performed using Prism 9 software (GraphPad Software, La Jolla, CA, USA). Statistical significance was set at *p* < 0.05.

## 3. Results

### 3.1. 6-OHDA-Responsive Expression of CLIC2 mRNA in Striatal Tissue

Microglial activation is a common feature of Parkinson’s disease, and we reported huge microglial accumulation in substantia nigra reticulata in our earlier work using 6-OHDA [15]. Considering the driving effects of CLIC2 on microglial invasion activity, we prepared 6-OHDA-treated unilateral Parkinson’s disease models where stratal dopamine content was decreased on the ipsilateral side (*p* < 0.0001, contralateral side, Figure 1A). Interestingly, we found the mRNA expression of CLIC2 was increased on the ipsilateral side (*p* < 0.001 vs. contralateral side, Figure 1C). Simultaneously, MMP-9 mRNA expression was also increased in similar fashion on the lesion side (*p* < 0.01 vs. contralateral side, Figure 1B).

### 3.2. 6-OHDA Increased Number of CLIC2-Expressing Cells in Substantia Nigra Pars Compacta

To identify the cell types expressing CLIC2 in the lesioned area of 6-OHDA-treated PD-model rat brain, triple immunofluorescence staining was performed using cryo-sections of the brain section at the substantia nigra pars compacta. Figure 2A displays localization of CLIC2 (red) with TH, which was visualized with white color, and CD11b was visualized in green color in the brain tissue of the 6-OHDA-treated unilateral Parkinson’s disease rat model. Morphometric analysis showed an increase CLIC2^+^ area in the lesioned side as compared to the un-lesioned side (*p*  <  0.05, Figure 2B). In addition to this, in contrast with contralateral side, the number of CLIC2^+^ cells also remarkably increased on the ipsilateral side (*p* < 0.01, Figure 2C). Collectively, this immunofluorescence observation suggests the involvement of CLIC2 in microglial infiltration in the lesion area of PD brains.

### 3.3. CLIC2 Regulates MMP Expression in Microglial Cells

Our previous report demonstrated that siRNA-induced CLIC2 knockdown in meningioma cells causes increased MMP-2 expression [13]. To elucidate the connection between CLIC2 and microglia MMPs, we also performed the same experiment and found that the expression of MMP-2 and MMP-9 decreased following CLIC2 knockdown in primary microglial cell cultures (*p* < 0.01, vs. control siRNA, Figure 3C,D). Next, we took advantage of a wheat protein synthesis system in which highly aqueous, buffer-soluble, GST-tagged recombinant CLIC2 was synthesized [18]. Recombinant CLIC2 protein was added to primary cultured microglia and incubated overnight, where recombinant CLIC2 increased the mRNA expression of MMP-9 (*p* < 0.05, vs. control medium, Figure 3B) but did not affect MMP-2 expression (Figure 3A).

### 3.4. CLIC2 Contributes Microglial Release of MMP-9

Considering the silencing of CLIC2 and modulatory effects of CLIC2 on microglial expression of MMP-2 and MMP-9, we performed a zymography assay, which is a simple technique that is commonly used to study the activity of MMPs. For this experiment, we used the supernatant from the samples in which recombinant CLIC2 was added to microglia and incubated overnight. As expected, exposure to recombinant CLIC2 increased the release of microglial MMP-9 (*p* < 0.05 compared to the control media, Figure 4). We also performed the same assay with CLIC2 knockdown and control samples, where CLIC2 knockdown decreased the release of MMP-9 (*p* < 0.05 compared to the irr-siRNA, Figure 4).

### 3.5. CLIC2 Plays Important Role in Microglial Invasion

Microglia shape the synaptic environment in healthy and diseased states, where they use MMPs to degrade extracellular matrices [19]. We performed an invasion assay to confirm the contributory role of CLIC2 in the microglial release of MMP-9 [20]. As anticipated, CLIC2 knockdown decreased the number of invading microglial cells (*p* < 0.01 compared to the IRR-siRNA, Figure 5C,D). In addition, exposure to recombinant CLIC2 increased the number of invading microglial cells (*p* < 0.05 compared to the control; Figure 5A,B).

### 3.6. CLIC2 Plays Vital Role in Microglial Metabolic Programing

Under healthy and injured conditions, microglia in the brain possess a highly ramified motile state that constantly extends and retracts to survey the local environment, which requires metabolic programing [21]. Oxidative phosphorylation and glycolysis are the two main sources of ATP production for microglial surveillance [22]; here, we assumed that CLIC2 addition strengthens the mitochondrial membrane, resulting in increased transmission of the respiratory electron transport chain to increase the yield of ATP. To address this possibility, OCR, which symbolizes mitochondrial oxidative respiration, was performed, where addition of recombinant CLIC2 peptide for 24 h increased OCR for basal respiration (*p* < 0.05), maximal respiration (*p* < 0.05), proton leak (*p* < 0.05), and spare respiratory capacity (*p* < 0.01) (Figure 6B). These data demonstrate that CLIC2 is crucially involved in microglial energy metabolism by preserving the standard functions of microglial mitochondrial oxidative phosphorylation.

## 4. Discussion

Microglia are highly dynamic cells that display functional, structural, transcriptional, and metabolic changes in response to both healthy and pathological alterations. Microglial cells are thought to be involved in the progressive loss of dopaminergic neurons in PD through the release of potentially harmful substances. These include pro-inflammatory cytokines and reactive oxygen/nitrogen species, which deleteriously affect neurons [1,17]. The expression of CLIC1 and CLIC4 in the microglia has been reported previously [11,23]. However, in this study, we provide the first evidence that healthy microglial cells express CLIC2 and that its expression is increased in the lesion area of PD-model rats. In this study, CLIC2 showed stimulatory effects on the microglial release of MMP-9 and its invasion capability. These data suggest that CLIC2 is an important regulatory component of microglial function.

The microglia regulate perineural and synaptic integrity in both healthy and diseased conditions to constantly sense perineural network and parasynaptic extracellular matrix integrity. Extracellular matrix remodeling has been widely documented in disease conditions [19], and recent research has demonstrated such remodeling in healthy homeostatic brains [19]. Microglial surveillance includes processes such as invasion, migration, and substrate degradation that recruit matrix-degrading enzymes. Previous reports have shown that microglia express heparinase as well as several MMPs and cathepsins [24]. Among MMPs, microglia mostly express MMP-2, MMP-9, MMP-12, and MMP-14, and only MMP-9 is predominantly expressed in these cells [20]. MMPs play vital roles in normal physiological activities of cells, such as morphogenesis, cell migration, and angiogenesis. MMPs are also involved in pathophysiological processes and have garnered attention in PD. Compared to age-matched controls, increased MMP-2 activity has been documented in postmortem brain tissue from patients with PD [25]. In the current study, we observed increased MMP-9 mRNA expression in a rat model of PD. In neuro-inflammatory conditions, MMPs play an important role (1) by activating enzymes for inflammatory molecules, (2) as signaling molecules, (3) in enhancing neurotoxicity, and (4) in compromising neurovascular integrity [26]. Here, we noticed that changes in microglial expression and the release of MMP-9 positively correlated with microglial transmigratory activity. A similar correlation was also reported in our previous work with meningioma cells, where MMP-2 expression and cellular invasion were suppressed by the addition of recombinant CLIC2 [13]. In addition, such effects were inversely reported in our recent report on microglia, suggesting a distinct role for CLIC2 in immune and tumor cells.

Sustained microglial functions require intracellular metabolic homeostasis, which involves a metabolic switch. A previous study reported that the addition of either glutamine or glucose increased microglial mitochondrial oxidative phosphorylation, which may be linked to microglial metabolic flexibility for supporting the surveillance of brain parenchyma [21]. Overall, integral mitochondrial oxidative phosphorylation is required for microglial function [27]. Inhibition of oxidative phosphorylation leads to the abolishment of microglial immune responses such as phagocytosis [28]. A study of NHE1 knockout microglia revealed higher mitochondrial oxidative phosphorylation, which is linked to the elevation of microglial phagocytic activity and enhanced synaptic remodeling [29]. The OCR evaluates the flow of electrons through the respiratory chain and all processes within the cell that can consume energy [30]. In this study, we found enhanced mitochondrial metabolism in microglia following 24-h incubation with recombinant CLIC2. Our findings identified the CLIC2 protein as a modulator for microglial immunometabolism and microglial MMP-9 activity to increase microglial invasion and surveillance. The current findings provide important information in neuroinflammatory disease, as it recognizes CLIC2 as a modulator of microglial invasion and immunometabolism.

There are some limitations of present study, as we were unable to confirm mechanisms underlying CLIC2 regulation on microglial MMP-9 activity and mitochondrial oxidative phosphorylation. In addition to this, astrocyte activation was accompanied by microglial activation [31,32], but here, we only concentrated on microglia.

## 5. Conclusions

As illustrated in Figure 7, our findings provide new evidence regarding the cellular and subcellular localization of CLIC2 in the substantial nigra pars compacta of rat PD-model brain tissue, and its expression increased following 6-OHDA administration. This study also showed that CLIC2 increased microglial oxidative phosphorylation and released MMP-9, which enhanced microglial invasion activity. Manipulation of microglial infiltration in PD brains through targeting CLIC2 may provide a new therapeutic target to prevent progression of this disease. However, the current findings warrant further studies to confirm the cellular mechanism of CLIC2 in the pathogenesis of the PD brain by using CLIC2 knockout rats.

## Figures and Tables

**Figure 1 brainsci-13-00055-f001:**
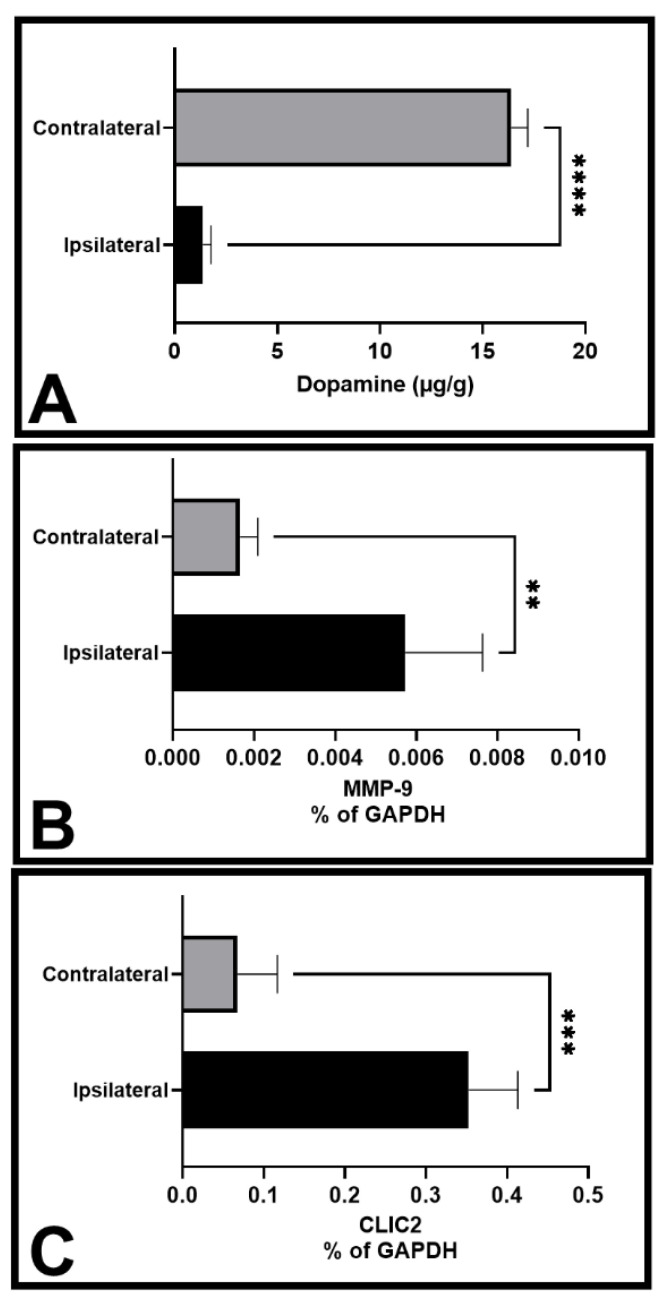
Expressional changes of CLIC2 and MMP-9 in Parkinson’s-modeled rat brain. Infusion of 6-OHDA decreased the contents of dopamine (DA) in striatum (**A**) and increased mRNA expression of MMP-9 (**B**) and CLIC2 (**C**). ** *p* < 0.01, *** *p* < 0.001, and **** *p* < 0.0001 indicate significant difference between the two ipsilateral and contralateral sides.

**Figure 2 brainsci-13-00055-f002:**
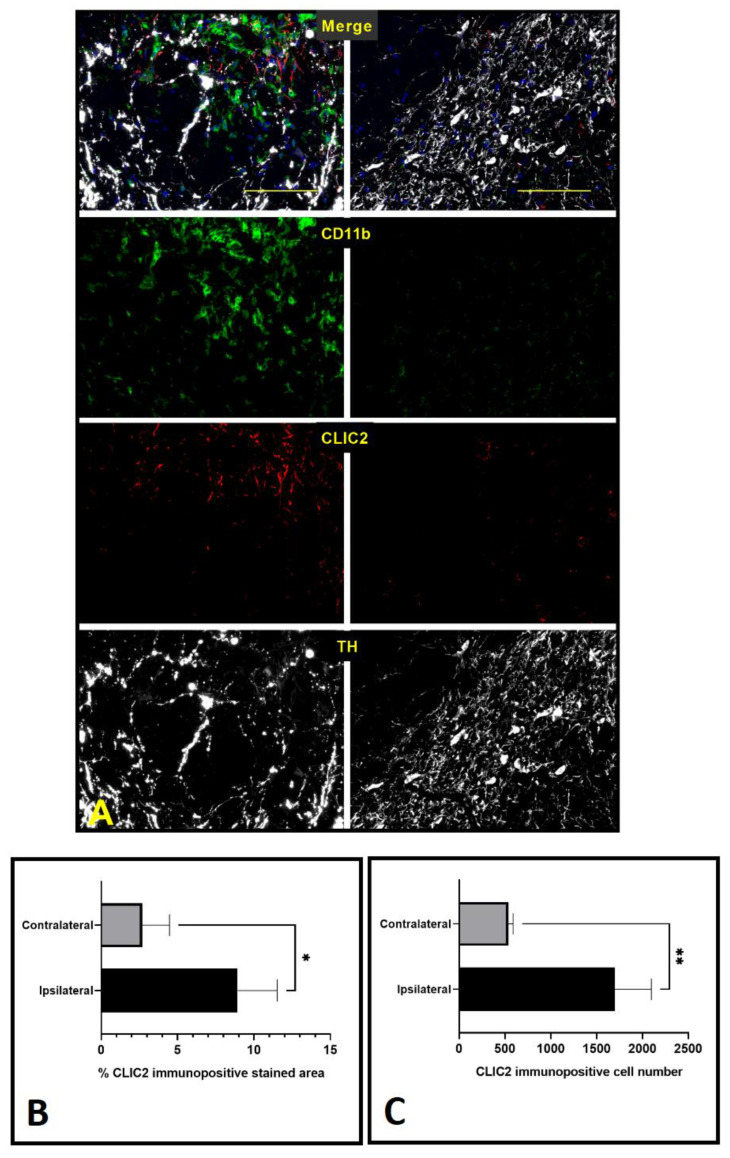
Colocalization of CLIC2 on microglia and dopaminergic neurons at the substantia nigra pars compacta of PD-modeled rat brain. Representative micrograph with Sale bar 100 μM where CLIC2 expressions on different cells were visualized with green color, neurons were marked with white color, and microglia were identified with red color (**A**), CLIC2 immunopositive stained area (**B**), CLIC2 immunopositive cell number (**C**). Data (*n* = 3) were presented as mean ± SD and analyzed with unpaired two-tailed *t*-test. * *p* < 0.05 and ** *p* < 0.01 indicate significant difference between ipsilateral and contralateral side.

**Figure 3 brainsci-13-00055-f003:**
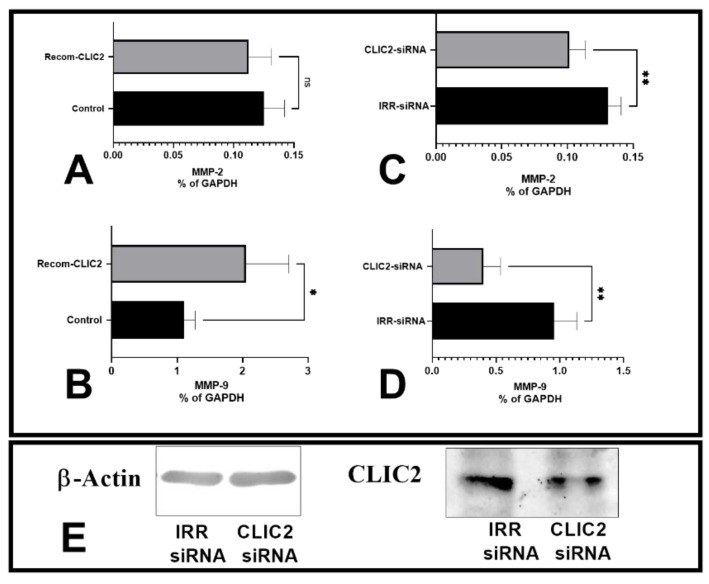
Effect of CLIC2 on microglial MMP expression. When primary cultured microglial cells were exposed to recombinant CLIC2, mRNA expression for MMP-2 and MMP-9 was increased (**A**,**B**). Additionally, using siRNA, CLIC2 expression were decreased, and concomitantly, gene expression of MMP-2 and MMP-9 decreased (**C**,**D**). Immunoblot confirming accuracy of CLIC2 siRNA knockdown experiments (**E**). Data (*n* = 4) were presented as mean ± SD and analyzed with unpaired two-tailed *t*-test. * *p* < 0.05, and ** *p* < 0.01 indicate significant difference, and (ns) *p* > 0.05 indicates nonsignificant difference between the two groups.

**Figure 4 brainsci-13-00055-f004:**
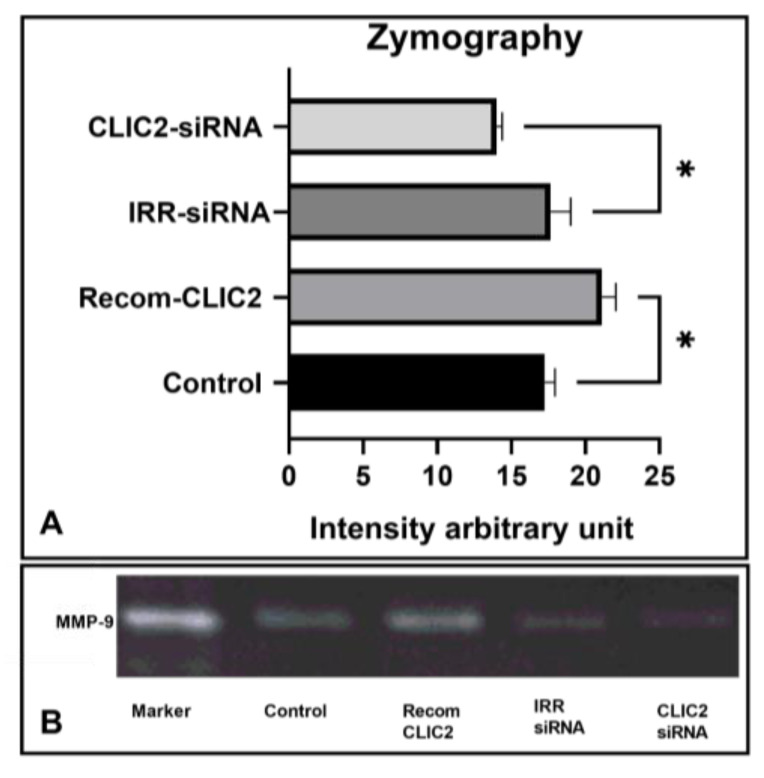
In microglia, CLIC2 increased the activity of MMP-9. Exposure of recombinant CLIC2 was found to increase MMP-9 activity, revealed by increased gelatinolysis in the zymographic assay using medium of primary cultured microglia, where siRNA-induced CLIC2 abolished microglia MMP-9 activity. Densitometric analysis of data (*n* = 3) was presented as mean ± SD and analyzed with unpaired two-tailed *t*-test. * *p* < 0.05 indicates significant differences between the two groups (**A**). Representative image data (**B**).

**Figure 5 brainsci-13-00055-f005:**
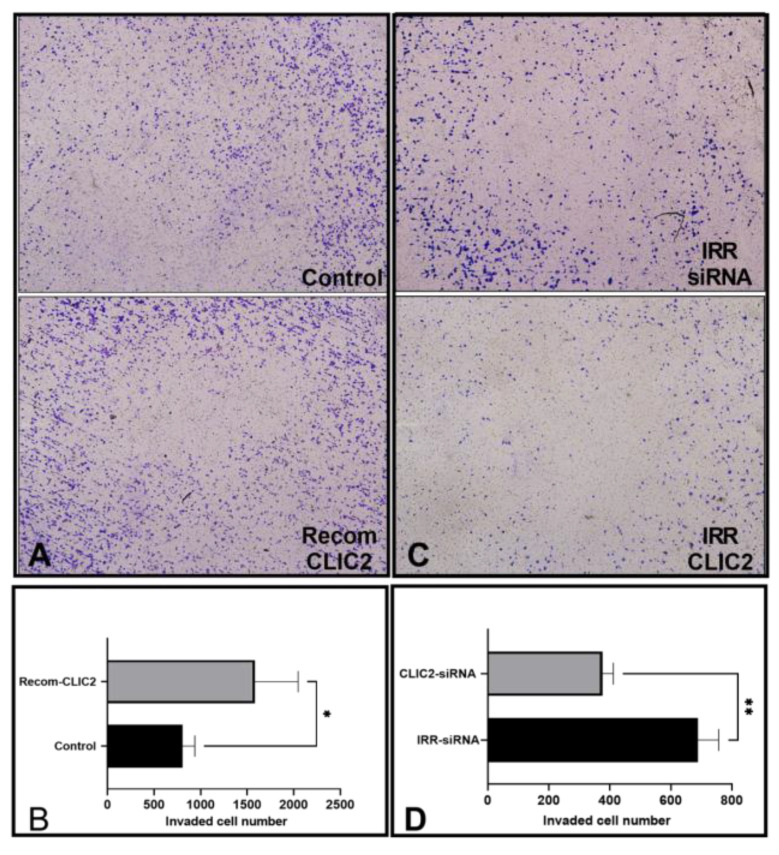
Modulatory effects of CLIC2 on microglial invasion activity. Recombinant CLIC2 accelerated the invasion of microglial cells. *n* = 4. CLIC2 silencing decreased the invasive activity of microglia (*n* = 3). Data are expressed as mean ± SD. Unpaired two-tailed *t*-test were applied. * *p* < 0.05 and ** *p* < 0.01 indicate significant differences between the two groups (**B**,**D**). Representative photographs for each group (**A**,**C**).

**Figure 6 brainsci-13-00055-f006:**
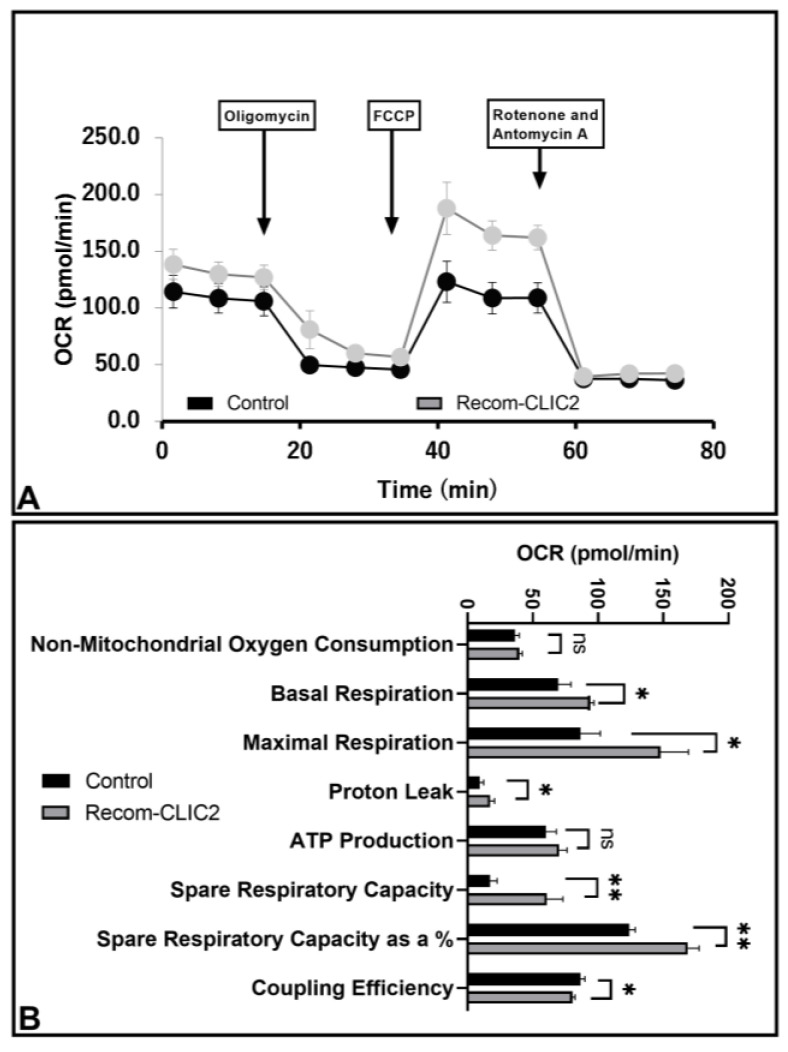
Modulatory effects of CLIC2 on the mitochondrial function in microglia. Recombinant CLIC2 exposure significantly enhanced mitochondrial function in primary cultured microglial cells. Mitochondrial oxygen consumption rate (OCR) (**A**) and mitochondrial extracellular acidification rate (ECAR) (**B**) were determined using a Seahorse XFp extracellular Flux Analyzer in microglia exposed to CLIC2 for 24 h. * *p* < 0.05 and ** *p* < 0.01 specify significant difference between the two groups.

**Figure 7 brainsci-13-00055-f007:**
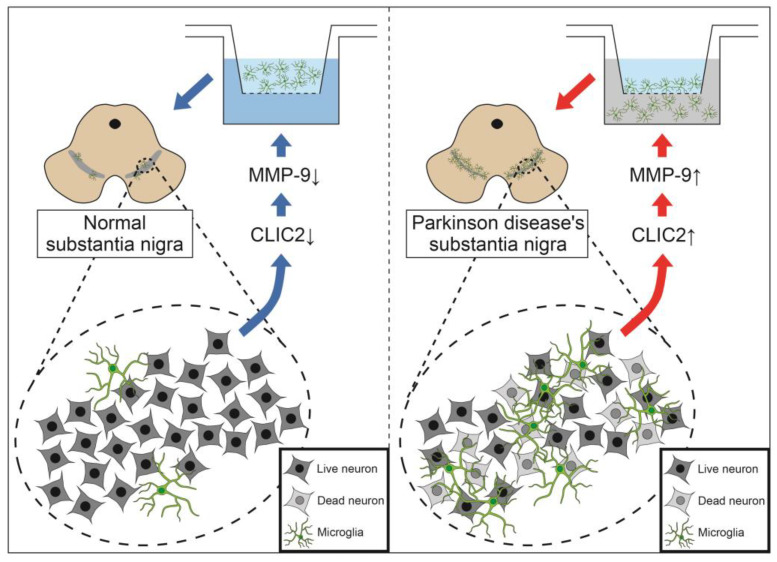
Diagram of the present hypothesis. Under normal condition (**Left**), CLIC2, MPP-9 expression, and microglial invasion in substantia nigra pars compacta are minimal. In PD condition (**Right**), increased CLIC2 expression results in increased MMP-9 activity and microglial invasion in the substantia nigra pars compacta.

## Data Availability

All data are available in the main text.

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
