# Peer review of "Chloride Intracellular Channel Protein 2 Promotes Microglial Invasion: A Link to Microgliosis in the Parkinson’s Disease Brain"

_brainsci, 2022, doi:10.3390/brainsci13010055_

Round 1

Reviewer 1 Report

This article cannot be published in this form. It is necessary to work on the structure of the data presentation, the authors should focus on a clear discussion of the results. The title of the article should reflect both the model of Parkinson's disease and the role of microglia on mitochondrial functions. In general, the title does not reflect the essence of the article. After processing, I'm ready to reconsider my decision

1) Figure 1 with immunocytochemical staining is not acceptable. Two-dimensional images should be presented showing the morphology of microglia and neurons. I recommend applying phase contrast. Readers should see culture density, microglia ratio, and other cell types. How the issue with the identification of astrocytes was resolved.

2) The results contain too many references to the work of other authors. References are appropriate to use in the chapter discussion of the results. I would recommend that the authors submit a separate discussion chapter of the results.

3) There is no conclusions section

4) In general, the results should be divided into subheadings, since the authors consider different effects by different methods.

Author Response

We are thankful to you for your valuable comments for this manuscript.

Here, we attached our responses where your comments are shown in blue font and our responses are in black font.

Reviewer 2 Report

In my opinion, the abstract is too general, and does not state the purpose of the study, moreover it  should include a synthetic description of: the research models used (primary microglia culture and PD model-6-OHDA-induced lesion in rat brain) and all the results obtained.

Please standardize the spelling: metallopeptidase (title) or metalloproteinase (text)

The section 2 Materials and methods:

No description of the methods whose results are presented in Fig. 5 and Fig. 6A

Fig. 1 The photomicrograph has poor quality, it should looks like in the publication ([17], Fig. 4)

Line 160 – it should be Figure 1B

The first sentence in paragraph 3.1 should be moved to discussion.

Line 160 instead of Figure 4B it should be Figure 1B

Fig. 2 The graphs show results for MMP-2 (A) and MMP-9 (B) proteins while the figure caption shows MMP-14 (line 182), additionally only MMP-9 expression is higher than control? - please correct

spelling: CLIC2 or Clic2; MMP9 or MMP-9?

Line 189:  it should be MMP-9 instead of MPP-9

Fig. 3 I think it will be better ”The effect of recombinant CLIC2 on the MMP-9 activity” (line 197) because figure relates only to this particular MMP.

Please explain the abbreviation „AU” on the Y axis

Fig. 4 Please correct the captions on the photomicrographs (invisible letters) and the description of the figure (line 213, 214 – A, B, C?), also please be more specific about how the cells are counted

Fig. 5 please explain the abbreviations (Oligo, FCCP, Rot/A) and what the red and blue line means?

Fig. 6 Line 244 – it should be MMP-9 instead of MPP-9

Line 246 – it should be (B and C)

Author Response

(The authors gave the same response as above.)

Reviewer 3 Report

The manuscript entitled ‘Regulatory role of chloride intracellular channel protein 2 on microglial matrix metallopeptidase 9 release while promoting microglial invasion’ by Choudhury et al, investigates the effects of the protein, chloride intracellular channel protein 2 (CLIC2) on microglial function and activation. They report that CLIC2 is expressed in microglial cells and regulate their activity partially through modulation and release of matrix metalloproteinase 9 (MMP-9). Subsequently, 6-OHDA mediated lesion of substantia nigra, a commonly used animal model of Parkinson’s disease increased CLIC2 and MMP-9 mRNA levels, indicative of enhanced microglial invasion activity. Overall, the paper addresses the important topic of studying proteins involved in microglial function, contributing to our understanding of their role in healthy and neuropathological conditions. Most of the manuscript is well written and crucial experiments have been conducted. However, the authors should address some of the following points to strengthen the paper.

1)    Abstract reads good, but authors should write the full form of MMP-9.

2) Introduction is well written, but authors should write a few sentences mentioning general functions of MMPs.

3)   Manuscript doesn’t mention gender of rats used. Please specify in Materials and Methods.

4)   Line 66: please specify what MFB is.

5)   In section 2.7: Statistics, authors don’t mention if data was tested for normality and any subsequent non-parametric tests used. Was this tested?

6)   There are several places in the manuscript where Figures and sub-panels have not been cited correctly in the Results section. Example: Line 160, 174, 178, 179 etc. Please correct these.

7)   For statistics in Results section, please write the t or F statistic along with exact p values.

8)  2-way ANOVA was specified to be used in the study but not mentioned anywhere in the manuscript. Can the authors clarify?

9)      Section 3.1: why was CLIC2 expression studied in prefrontal cortex sections?

10)  In Figure 1, please add scale to Fig 1 A & B.

11)  Section 3.2, retitle as CLIC2 ‘regulates’ MMP expression in microglial cells.

12)  In section 3.2, authors mention that CLIC2-siRNA decreased CLIC2 levels, but never show this. Can they add a graph showing this?

13)  In section 3.2, can the authors speculate on why Recom-Clic2 did not increase MMP-2 levels?

14)  In Figure 2, line 182: what is MMP-14 and why is it mentioned that both MMP-9 & 14 were increased?

15)  What is control in Fig 2A & B? Is it the same as Irr-SiRNA?

16)  Can the authors show pictures of CLIC2 and MMP-9 co-localizing in same microglial cells or cite papers showing the same, to support Figure 2 results?

17)  Please cite correct panels in the figure captions i.e line 213 & 214.

18)  In section 3.5, is the change in OCR and ECAR after addition of recombinant CLIC2 compared against a control or baseline? And statistics for these results are not mentioned.

19)  Please specify what the blue and red lines indicate in Fig 5?

20)  Please specify what Oligo, FCCP and Rot/A signify in Fig 5 A &B.

21)  In section 3.6, why was no control performed for injection of 6-OHDA i.e. injection of vehicle for 6-OHDA?

22)  In section 3.6, an experiment to test effects of CLIC2 siRNA on CLIC2 and MMP-9 levels and maybe restoration of DA levels in the 6-OHDA experiment, might be important but not required.

23)  In section 3.6, was dopamine measured in any specific subregion of the striatum? It is not mentioned in the Methods section, how dopamine was measured and how long did the authors wait for to measure dopamine levels and CLIC2 and MMP-9 mRNA levels after 6-OHDA injections. Please specify these details.

24)  Discussion section is well written, however authors should discuss/ speculate mechanisms underlying CLIC2 regulation of MMP-9 activity.

Author Response

(The authors gave the same response as above.)

Round 2

Reviewer 1 Report

The quality of the article has been significantly improved The authors should discuss in more detail the reactivation of astrocytes and microglia, taking into account the two main types of reactive astrogliosis (A1 and A2). https://pubmed.ncbi.nlm.nih.gov/22553043/

https://pubmed.ncbi.nlm.nih.gov/34884629/

The conclusion should be written in more detail.

Author Response

Thank you very much for the encouraging comments. Here, we attached our responses where your comments are shown in blue font and our responses are in black font.
